# Processed Meat Consumption and the Risk of Cancer: A Critical Evaluation of the Constraints of Current Evidence from Epidemiological Studies

**DOI:** 10.3390/nu13103601

**Published:** 2021-10-14

**Authors:** Mina Nicole Händel, Jeanett Friis Rohde, Ramune Jacobsen, Berit Lilienthal Heitmann

**Affiliations:** 1Research Unit for Dietary Studies, The Parker Institute, Bispebjerg and Frederiksberg Hospital, 2000 Frederiksberg, Denmark; mina.nicole.holmgaard.handel@regionh.dk (M.N.H.); jeanett.friis.rohde@regionh.dk (J.F.R.); 2Research Group for Social and Clinical Pharmacy, Department of Pharmacy, University of Copenhagen, 2100 Copenhagen, Denmark; ramune.jacobsen@sund.ku.dk; 3Section for General Practice, Department of Public Health, University of Copenhagen, 1014 Copenhagen, Denmark

**Keywords:** processed meat, cancer, systematic review, meta-analysis, GRADE, AMSTAR, ROBINS-I, dietary guidelines

## Abstract

Based on a large volume of observational scientific studies and many summary papers, a high consumption of meat and processed meat products has been suggested to have a harmful effect on human health. These results have led guideline panels worldwide to recommend to the general population a reduced consumption of processed meat and meat products, with the overarching aim of lowering disease risk, especially of cancer. We revisited and updated the evidence base, evaluating the methodological quality and the certainty of estimates in the published systematic reviews and meta-analyses that examined the association between processed meat consumption and the risk of cancer at different sites across the body, as well as the overall risk of cancer mortality. We further explored if discrepancies in study designs and risks of bias could explain the heterogeneity observed in meta-analyses. In summary, there are severe methodological limitations to the majority of the previously published systematic reviews and meta-analyses that examined the consumption of processed meat and the risk of cancer. Many lacked the proper assessment of the methodological quality of the primary studies they included, or the literature searches did not fulfill the methodological standards needed in order to be systematic and transparent. The primary studies included in the reviews had a potential risk for the misclassification of exposure, a serious risk of bias due to confounding, a moderate to serious risk of bias due to missing data, and/or a moderate to serious risk of selection of the reported results. All these factors may have potentially led to the overestimation of the risk related to processed meat intake across all cancer outcomes. Thus, with the aim of lowering the risk of cancer, the recommendation to reduce the consumption of processed meat and meat products in the general population seems to be based on evidence that is not methodologically strong.

## 1. Introduction

Both the production and consumption of red meat and preserved or processed meat products (defined as meats that have undergone changes, i.e., salting, curing, smoking, or adding chemical preservatives) have been rapidly increasing over recent decades, most significantly in emerging economies [1]. In addition to total energy intake, meat is an essential source of protein, fat and fatty acids, and essential micronutrients, for example, heme iron, selenium, choline, vitamin B6, thiamine, niacin, and riboflavin. However, due to several components that arise from the processes of cooking or processing meat, such as polycyclic aromatic hydrocarbons, advanced glycation end products, and heterocyclic amines, as well as sodium/salt, nitrite, nitrate, and nitrosamines, a high consumption of meat and processed meat products has been suggested to have severe detrimental effects on the health of humans, including the risk of cancer [2].

Under the auspices of the World Health Organization [3], the International Agency for Research on Cancer (IARC), an independent cancer agency, has been coordinating with the European Commission to prepare the European Code Against Cancer, which includes 12 ways to reduce cancer risk [4]. In an effort to inform the public about reducing cancer risk, the 2012–2013 edition of the code recommended avoiding processed meat while also limiting the consumption of red meat and foods high in salt. In 2018, IARC summarized that there is now “sufficient evidence in humans for the carcinogenicity of consumption of processed meat. Consumption of processed meat causes cancer of the colorectum. Positive associations have been observed between consumption of processed meat and cancer of the stomach” [5]. The IARC Monograph also included a statement that red meat consumption was “probably carcinogenic” because bias and confounding could not be ruled out, yet failed to acknowledge that the same studies, and usually the same publications, reported on both red and processed meat intake with identical methods. Therefore, the processed meat studies must have been subject to the same limiting factors. The World Cancer Research Fund (WCRF), which included some of the same members from the IARC working group, also reported in an update to the WCFR evidence paper that a high intake of processed meat was associated with a high risk of colorectal cancer (CRC) [6]. The latest U.S. Dietary Guidelines for Americans (DGA), released in late 2020, does not include a top-level recommendation to reduce red or processed meats, yet the DGA has long focused on choosing “lean meat” due to its lower saturated fat content. In 2015, the DGA began to focus on dietary patterns rather than nutrient-based recommendations [7], and the current 2020–2025 DGA [8] states several times that “common characteristics of dietary patterns associated with positive health outcomes” include a ”relatively lower consumption of red and processed meats”. The systematic reviews for the 2020 DGA [9] concluded that there was “moderate” evidence for recommending one of the DGA’s three “healthy dietary patterns” to protect against breast and colorectal cancer and “limited” evidence for protecting against lung and prostate cancer. The “moderate” conclusions for breast and colorectal cancer are based on reviews that cite 1–2 randomized controlled clinical trials (RCTs) [10,11,12]. Systematic reviews that specifically analyze the effects of red and processed meat and cancer outcomes have never been conducted for the U.S. DGA [13]. Reviews have instead looked collectively at “animal protein products”, including eggs, fish, and dairy, and, therefore, have not isolated the health effects of red or processed meat. Similarly, dietary guidelines in Europe, that is, the United Kingdom [14] and Scandinavia [15], also recommend that the intake of both red and processed meats should be limited.

To date, few reviews report only relative effects of red and processed meat on cancer outcomes, and few reviews—if any—report absolute effects. While relative effects for red and processed meat may be positive and statistically significant, absolute effects are small (less than 1%) [16]. Further, dietary guidelines rarely, if ever, consider public values and preferences. Thus, while reductions in meat consumption are clearly advisable for sustainability and environmental concerns, public willingness to modify red and processed meat consumption may be less likely based on small and uncertain health effects [16].

## 2. Methodological Limitations of Systematic Reviews on Processed Meat and the Risk of Cancer

Until now, there have been few randomized trials that have investigated the consumption of red meat and the risk of colon cancer, as recently reviewed by Johnston and colleagues [17]. Similarly, only two trials have examined the effect of different dietary patterns on cancer risk, only one of which was red meat intake [10,18], and both of which showed significant reductions in meat did not change cancer risk [10,18]. On the other hand, there is a large volume of observational studies, in total, 31 prospective cohort studies that include data from 3.5 million participants [17] and many more case-control studies that have examined if cancer patients recalled a different previous processed meat intake than non-cancer cases. There are more than one hundred summary papers that have reviewed and performed meta-analyses based on these primary studies [19], which exceeds the number of original studies by far.

In a recent overview published in 2019 [19], we conducted a thorough, systematic assessment of the general methodological quality of these systematic reviews of processed meat only using the AMSTAR criteria [20,21]. AMSTAR stands for A MeaSurement Tool to Assess Systematic Reviews. This is a valid, reliable, and widely used measurement instrument that helps researchers differentiate between systematic reviews, focusing on their methodological quality. The quality can be categorized as high, moderate, or low.

We used the Grading of Recommendations Assessment, Development, and Evaluation (GRADE) approach [22] to assess the strength of recommendation in order to evaluate the certainty of the estimates of individual outcomes from the published systematic reviews and meta-analyses [19] on processed meat consumption and the risk of chronic disease morbidity and mortality, including cancer at different sites across the body, as well as the overall risk of cancer mortality. GRADE provides a reproducible and transparent framework for grading the certainty of evidence with four levels of certainty: very low, low, moderate, and high. For each of GRADE’s five domains assessed for each study (risk of bias, imprecision, inconsistency, indirectness, and publication bias), the review authors have the option of decreasing their level of certainty by one or two levels. For observational studies, there is also the possibility of increasing the level of certainty by one or two levels if there is a large magnitude of effect, a strong dose-response gradient, or plausibility that residual confounding would further support inferences regarding an effect. We further explored if discrepancies in study designs and risks of bias could explain the heterogeneity observed in meta-analyses.

Studies had to comply with the following two main quality requirements (two of the items in AMSTAR) to be included in our review [19]: (1) they must have documented a quality assessment of the primary studies, with no restriction on the quality assessment tool, and (2) they must have performed a comprehensive literature search, defined as a search performed in at least two databases relevant to the research question. More than 100 reviews were excluded because they had not performed a quality assessment on the primary studies included in the review. In total, only 22 of 130 reviews and meta-analyses met these two basic criteria and were subsequently included in our overview of reviews. Of the 22 reviews, 19 reported on cancer outcomes (the other outcomes were type 2 diabetes and cardiovascular disease). According to our AMSTAR evaluation, these 19 cancer reviews were generally only of moderate methodological quality, and the methodological quality in the reviews do not improve with time (Figure 1), despite several attempts to improve the reporting of systematic reviews and meta-analyses already in the 2000s, for instance, with AMSTAR [20,21], Preferred Reporting Items for Systematic Reviews and Meta-Analyses (PRISMA) [23], and GRADE [22].

The main identified methodological shortcomings were (1) a lack of a reference to a predetermined/a priori published research objective, that is, a protocol or an ethics approval, which, according to AMSTAR, indicates a high risk of selectively reported results; (2) incomprehensive literature searches, which indicates a high risk of overlooking relevant literature; (3) not considering the scientific quality of the evidence in formulating the conclusions, which indicates a high risk of emphasizing results from weak study designs; (4) not reporting the conflicts of interest of the authors of the reviews as well as those of the original included primary studies.

Our results indicate that all the reviews and meta-results that were based on case-control studies (Figure 1), which, by their nature, are retrospective and are, therefore, prone to the misclassification of exposure in relation to processed meat consumption, were likely to overestimate the risk of having cancer. A high consumption of processed meat was generally associated with a risk of cancer in the digestive system, including the esophagus, stomach, colorectum, and pancreas, but the results differed greatly according to whether they came from case-control or cohort studies. Generally, cancer risk seemed to be higher in case-control studies than in cohort studies, which may suggest that the better prospective study designs generally gave less evidence for an association. Due to the well-known methodological limitations of case-control studies, such as information bias, and the established fact that people are not able to remember accurately what they have eaten in the past, results based on case-control studies should be interpreted cautiously. The findings for an association between processed meat intake and cancer of the digestive system spanned from a higher risk of approximately 30–70% in the case-control studies [24,25,26,27,28,29], to a very modest or no association in the results of the meta-analyses that exclusively examined cohort studies [24,25,26,27,28,29,30].

For other cancers, often only case-control studies were available. For instance, the risk of cancer of the oral cavity and oropharynx was 91% higher among the cases that reported having had a higher consumption of processed meat compared to controls [31]. The results of this meta-analysis included nine case-control studies (cases: *n* = 4104, controls: *n* = 501,730).

For head and neck cancer (nasopharyngeal carcinoma), the risk was 46% higher among cases with a processed meat intake below 30 g/week compared to those who reported never eating processed meat [32]. These meta-analysis results were based on 13 case-control studies, including 5849 cases and 12,735 controls.

Only a modestly higher risk among high compared to low consumers of processed meat was seen in relation to non-Hodgkin lymphoma (17%), renal cell carcinoma (13%), and overall cancer mortality (13%). For non-Hodgkin lymphoma and renal cell carcinoma, the results were based on a mixture of case-control and cohort studies, while overall cancer mortality was based solely on cohort studies. Processed meat consumption did not seem to be associated with cancer in the liver, brain (glioma), ovaries, or lung [33,34,35,36,37].

## 3. Methodological Limitations of the Primary Studies on Processed Meat and Cancer

In 2020 [38], we performed a meta-analysis in which we investigated the association between processed meat and the risk of CRC, colon, and rectal cancer, and we thoroughly evaluated the quality of the original studies. The quality assessment was undertaken using Cochrane’s Risk Of Bias In Non-randomized Studies of Interventions (ROBINS-I) assessment tool [39], by which the risk of bias is assessed within seven different areas of methods applied to observational studies and is an instrument similar to the one scientists use when evaluating the risk of bias in clinical trials. Such an evaluation provided us with new insights into the internal validity of the reviews and meta-analyses included in our overview of reviews [19]. For the meta-analysis [38], we included 29 observational prospective cohort studies conducted from 1990 to 2015 in Europe, Australia, Asia, and North America. The results are similar to previously reported estimates from meta-analyses of cohort studies [28,40,41,42,43,44,45,46,47], with a 13% higher risk of CRC, a 19% higher risk of colon cancer, and a 21% higher risk of rectal cancer among those with the highest processed meat intake. We concluded that due to the risk of bias, especially from confounding and missing data and selective outcome reporting, the possibility could not be excluded that these associations were distorted and could be either over- or underestimated [38].

Using the GRADE approach, we concluded that the overall certainty for the body of evidence examining the association between processed meat and cancer was very low across all individual cancer outcomes, meaning that the true effect could be markedly different from the estimated effect [19,38]. Our reason for rating down our certainty in these studies was due to the serious risk of bias (issues regarding confounding, missing data, and the risk of selection of the reported results were not sufficiently addressed), serious imprecision due to wide confidence intervals, and serious inconsistency due to unexplained variability between the included studies (so-called heterogeneity) [19,38]. Indirectness or publication bias were not issues in this research field [19,38].

The rationale for the GRADE evaluation (very low certainty of the effect estimates) was, first, that the results were based exclusively on observational studies, many of which were of retrospective case-control design, and which, by default, are considered low quality in the GRADE approach. Theoretically, observational studies can be upgraded to moderate quality if there is a large effect size or a strong dose-response relationship, but these criteria were not met for any of the included results.

Secondly, we considered whether the exposure (processed meat) was measured accurately. Using our updated meta-analysis of CRC as an illustration of what we assume is representative across cancer outcomes [38], we could see that the definition of processed meat varied greatly among studies. Processed meat was either classified by referring to the preservation methodology, by listing individual food items, or with no further definition. In addition, processed meat was often ascertained using validated food frequency questionnaires (FFQ). In general, FFQs perform almost as well as 7-day weighed diet records [48], and although they have some advantages because they can be administered repeatedly during follow-up to account for changes in diet over time, they are like other diet instruments prone to some misclassification. While it is possible that FFQs do not give reliable results over multiple administrations, repeated applications of FFQs are rarely done, and the results from the few studies that have done so suggest that the diet is not stable over time [49,50]. FFQ data are further challenged when subjects are required to recollect their food consumption from up to 10 years ago [51] with the use of an incomplete food list, the inability to give complete information on portion sizes, the inability to give complete information on cooking practices, and so forth. [38].

Third, because assignment to high or low processed meat consumption is not random, as it would be in trials, we considered if there had been appropriate control for confounding (factors that both influence processed meat intake and cancer outcomes, such as age, sex, family history of CRC, BMI/overweight, energy intake, alcohol, and smoking), including those that are unmeasured or might involve time-varying confounding. Even in the most well-conducted prospective observational studies, unobserved or residual confounding can still be present, and known confounders may still be measured imprecisely and/or using non-validated methods. In our updated meta-analysis on CRC [38], all but two of the eligible studies failed to control for age, sex, family history of CRC, BMI/overweight, energy intake, alcohol, and smoking. These were the prespecified confounders for which the eligible studies were obliged to control in order to receive a low risk of bias in the ROBINS-I tool. This problem was commented on by Gong et al. (2020) [52] in response to the recent *Guideline to recommending on unprocessed red meat and processed meat consumption* by Johnston and colleagues [17], in which Gong et al. calculated a so-called E-value analysis to demonstrate how strong any unmeasured confounding would have needed to be to negate the observed results. For all outcomes assessed, including CRC, none had an E-value upper confidence interval greater than 2.5, implying that an unobserved confounder is 2.5 times more likely to be associated with the studies on cancer type. This means that the suggested association between processed meat consumption and adverse cancer outcomes does not seem very robust and may potentially not be causal because it is highly possible that the observed association would be nullified if the unobserved confounder had been included in the statistical model.

Finally, considering the loss to follow-up (the risk of bias associated with missing data) and selective outcome reporting, our updated meta-analysis of CRC indicated that 75% of the eligible studies had a moderate to serious risk of missing data, and about half of the studies had issues with bias in the selection of the reported results.

Limitations to the GRADE approach in evidence of diet and health, such as processed red meat and cancer, have been proposed by Qian et al. [53]. Since it may be infeasible to conduct informative (long-term) randomized trials ensuring blinding or to show strong dose-response relationships, the conclusion of low certainty evidence may be inevitable [53]. Instead, Qian et al. suggested that observational studies should be upgraded if they fulfill several of the ten Bradford Hill criteria: strong association/effect, consistent findings, temporality (cause precedes effect), dose–response relationship, plausibility, coherence between epidemiological and laboratory findings, reversibility (if the cause is deleted then the effect would disappear as well), experiment (experimental evidence enhances the probability of causation), and analogy (existing similar associations would support causation). However, most of the Bradford Hill criteria are already embedded in GRADE, as described by Schünemann et al. more than a decade ago [54]. We do acknowledge that the different types of study designs within observational studies are not well captured in the rating by GRADE. Therefore, considerations about what type of study designs that best address the research question should be given high priority in the initial phases of conducting a systematic review [55]. In summary, there are severe methodological limitations to the majority of the previously published systematic reviews and meta-analyses linking processed meat to cancer risk. They generally lacked a proper risk of bias assessment of the primary studies included, and it seemed that the literature searches may have been selective in some instances. In the primary studies, there were potential consequences for the misclassification of exposure, a serious risk of bias due to confounding, a moderate to serious risk of bias due to missing data, and a moderate to serious risk of selection of the reported results, all of which may have led to the overestimation of associations with all cancer outcomes. Hence, the findings of a causal relationship between processed meat and cancer in both reviews and primary studies are suspected to be associated with uncertainty [19,38]. This finding is supported by the recent results from Johnston et al. [17], whose systematic review was also based on GRADE and reached a similar conclusion and provided new guidelines for the intake of processed meat.

Thus, the recommendation to reduce the consumption of processed meat and meat products to protect against cancer in the general population does not seem to be convincingly substantiated on the evidence that is methodologically strong. Clearly, there is still a lack of randomized trials evaluating the effect of lowering processed meat intake, and while such trials may be infeasible, cohort studies do not lend strong support for an association.

## Figures and Tables

**Figure 1 nutrients-13-03601-f001:**
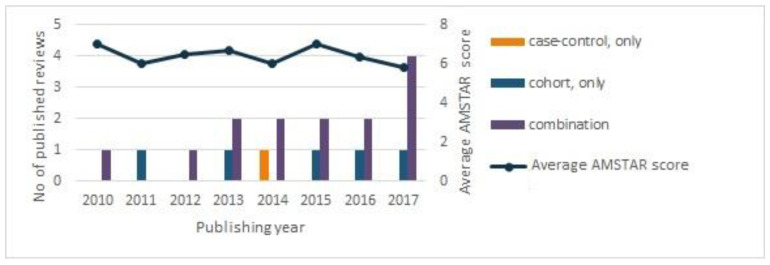
Overview of the number of published systematic reviews and the average AMSTAR (A MeaSurement Tool to Assess Systematic Reviews) score in the systematic reviews according to publishing year.

## Data Availability

Data is available upon request.

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
