# Peer review of "Processed Meat Consumption and the Risk of Cancer: A Critical Evaluation of the Constraints of Current Evidence from Epidemiological Studies"

_nutrients, 2021, doi:10.3390/nu13103601_

Round 1

Reviewer 1 Report

This manuscript presents a well-reasoned explanation why observational studies do not support strong conclusions regarding diet and disease. There are a few minor issues that the authors should clarify.

Page 2, paragraph 2, last sentence, the IARC Monograph included a statement that red meat consumption was class 2, probably carcinogenic, because bias and confounding could not be ruled out, yet failed to acknowledge that the same studies, and usually the same publications, reported on both red and processed meat intake with identical methods. Therefore, the processed meat studies must have been subject to the same limiting factors.

Page 5, paragraph 1, line 9, repeated application of FFQs is rarely done and, to my knowledge, only regularly done with the Harvard University cohorts. But the only published data on diet over time is in WC Willett et al, New Engl J Med, 340:169, 1999, who reported that for dietary fiber,  77% of women changed 2 or more quintiles over a 6-year period, suggesting that either diet is not stable over time or the FFQ does not give reliable results over multiple administrations. This is another limitation that should be acknowledged.

Page 5, paragraph 2, last sentence, suggest replacing increase with associate with, since this type of study cannot demonstrate causality unless the RR is strong and robust.  

Page 6, paragraph 1, line 6, typo in “considerations about what type ...”

Author Response

We thank the reviewers for their comments and suggestions which we believe have strengthened the manuscript further. We appreciate the opportunity to elaborate on the points raised by the reviewers.

Please find below our point-by-point replies to the reviewers’ comments, which are formatted in bold, followed by our reply in blue colored characters, and alterations highlighted in red.

Revisions in the manuscript are clearly highlighted, using the "Track Changes" function in Microsoft Word, so that changes are easily visible to the editors and reviewers.

Please be aware that the references in the manuscript provided by Nutrients were no longer “active”. So, the references are applied as comments.

Reviewer 1:

This manuscript presents a well-reasoned explanation why observational studies do not support strong conclusions regarding diet and disease. There are a few minor issues that the authors should clarify.

 Page 2, paragraph 2, last sentence, the IARC Monograph included a statement that red meat consumption was class 2, probably carcinogenic, because bias and confounding could not be ruled out, yet failed to acknowledge that the same studies, and usually the same publications, reported on both red and processed meat intake with identical methods. Therefore, the processed meat studies must have been subject to the same limiting factors.

 Thank you. We have added this information to the manuscript on page 3, paragraph 1, last sentence:

The IARC Monograph also included a statement that red meat consumption was “probably carcinogenic”, because bias and confounding could not be ruled out, yet failed to acknowledge that the same studies, and usually the same publications, reported on both red and processed meat intake with identical methods. Therefore, the processed meat studies must have been subject to the same limiting factors.

 Page 5, paragraph 1, line 9, repeated application of FFQs is rarely done and, to my knowledge, only regularly done with the Harvard University cohorts. But the only published data on diet over time is in WC Willett et al, New Engl J Med, 340:169, 1999, who reported that for dietary fiber,  77% of women changed 2 or more quintiles over a 6-year period, suggesting that either diet is not stable over time or the FFQ does not give reliable results over multiple administrations. This is another limitation that should be acknowledged.

Thank you for this comment. We have now acknowledged this limitation in the manuscript on page 6 paragraph 1, line 10:

While it is possible that FFQs do not give reliable results over multiple administrations, repeated applications of FFQs are rarely done, and the results from the few studies that have done so, suggest that the diet is not stable over time [references],    

Weismayer C, Anderson JG, Wolk A. Changes in the stability of dietary patterns in a study of middle-aged Swedish women. J Nutr. 2006;136:1582–7.

Mishra GD, McNaughton SA, Bramwell GD, Wadsworth ME. Longitudinal changes in dietary patterns during adult life. Br J Nutr. 2006;96:735–44.

 Page 5, paragraph 2, last sentence, suggest replacing increase with associate with, since this type of study cannot demonstrate causality unless the RR is strong and robust.

Thank you for this suggestion and although we agree, we have deleted this sentence due to a comment made by reviewer 2.

Page 6, paragraph 1, line 6, typo in “considerations about what type ...”

Thank you. The typo on line on Page 7, paragraph 3, line 6 has now been corrected: “considerations about what type ...”

Reviewer 2 Report

Händel and colleagues present an analysis of the body of evidence supporting recommendations to reduce processed meat consumption. The authors draw on an overview and a systematic review to conclude that reviews and primary studies addressing the health effects of processed meat consumption have critical methodological limitations.

The authors’ concerns are justified and well-supported. In fact, the authors are among a small cadre of methods researchers who have pointed to similar limitations in nutrition research and recommendations—particularly related to red and processed meat consumption.

While the arguments are justified, my only critical concern is whether this piece is sufficiently novel and presents anything beyond the authors’ overview and systematic review that are already published.

Major issues

1. In their description of their overview, the authors describe limitations in systematic reviews and limitations of primary studies. These are two separate issues. The manuscript will benefit from subsections that separate these issues. Additionally, the manuscript will benefit from subsections separating the discussion of the overview and the discussion of the systematic review.

2. “In a recent overview published in 2019, we conducted a thorough, systematic assessment of the general methodological quality of these systematic reviews using the AMSTAR criteria [19;20].”

Here, the authors should reference their overview. The authors should also clarify whether this overview addressed systematic reviews of processed meat only, red and processed meat, or general dietary exposures.

3. “More than 100 reviews were excluded, because they had not performed a quality assessment on the primary studies included in the review. In total, only 22 of 130 reviews and meta-analyses met these two basic criteria, and were subsequently included in our overview of reviews.”

Have these 22 reviews previously informed any dietary recommendations? It is possible that only the best systematic reviews are used to inform dietary recommendations. The authors should acknowledge this possibility.

If feasible, the authors may consider using a sample of dietary guidelines by authoritative organizations and government bodies and examining the quality of systematic reviews that informed recommendations on red and processed meat.

4. Two additional issues likely explain recommendations against processed meat. To date, few reviews only report relative effects and few reviews—if any—report absolute effects. While relative effects for red and processed meat may be positive and statistically significant, absolute effects are small. Further, dietary guidelines rarely, if ever, consider public values and preferences. If guideline panels considered values and preferences, they would likely realize that few participants would be willing to modify their consumption of red and processed meat based on such small and uncertain effects. A discussion of these two issues will strengthen the paper. See: Johnston BC, Guyatt GH. Causal inference, interpreting and communicating results on red and processed meat. Am J Clin Nutr. 2020 May 1;111(5):1107-1108.

Minor Issues

1. “Until now, there have been few randomized trials that investigated the consumption of red meat and risk of colon cancer, as recently reviewed by Johnston and colleagues [16], and similarly, only two trials examining the effect of processed meat intake on cancer risk in multi-factorial designs that also including other dietary changes [10;17], both of which showed significant reductions in meat, without changes in cancer risk [10;17].”

The trials that the authors describe as ‘multifactorial’ investigated the effects of interventions that only modified red meat consumption secondary to other characteristics of the diet. The Women’s Health Initiative tested a low-fat diet which resulted in reduced red meat consumption. The Polyp Prevention Trial tested a high fiber diet which also resulted in reduced red meat consumption. The trials were not factorial in design (defined as a trial that tests the effects of more than two interventions simultaneously). To avoid confusion with ‘factorial trials’, I suggest the authors avoid the term ‘multifactorial’ and instead describe that the interventions tested in the trials modified the diet of the participants in many ways, only one of which was red meat intake.

2. “Our reason for rating down our certainty in these studies was due to the serious risk of bias (issues regarding confounding, missing data, risk of selection of the reported results were not sufficiently addressed), serious risk of imprecision due to wide confidence intervals, and serious risk of inconsistency due to unexplained variability between the included studies (so-called heterogeneity), while there were no worrying issues regarding indirectness or publication bias [18;37].”

Please remove ‘risk of’ for imprecision and inconsistency.

3. “The issues on confounders are further complicated, as some of the statistical models include mediators, and/or colliders, and thus there is a high risk of over-adjustment bias, which may distort the result. For instance, red meat may increase weight gain and could act as a mediator that should not be controlled.”

This is not a compelling argument because over adjustment would typically reduce observed associations whereas analyses addressing the association between processed meat and adverse health outcomes are typically positive and statistically significant.

Author Response

We thank the reviewers for their comments and suggestions which we believe have strengthened the manuscript further. We appreciate the opportunity to elaborate on the points raised by the reviewers.

Please find below our point-by-point replies to the reviewers’ comments, which are formatted in bold, followed by our reply in blue colored characters, and alterations highlighted in red.

Revisions in the manuscript are clearly highlighted, using the "Track Changes" function in Microsoft Word, so that changes are easily visible to the editors and reviewers.

Please be aware that the references in the manuscript provided by Nutrients were no longer “active”. So, the references are applied as comments.

Reviewer 2:

Händel and colleagues present an analysis of the body of evidence supporting recommendations to reduce processed meat consumption. The authors draw on an overview and a systematic review to conclude that reviews and primary studies addressing the health effects of processed meat consumption have critical methodological limitations.

 The authors’ concerns are justified and well-supported. In fact, the authors are among a small cadre of methods researchers who have pointed to similar limitations in nutrition research and recommendations—particularly related to red and processed meat consumption.

 While the arguments are justified, my only critical concern is whether this piece is sufficiently novel and presents anything beyond the authors’ overview and systematic review that are already published.

 Major issues

 1. In their description of their overview, the authors describe limitations in systematic reviews and limitations of primary studies. These are two separate issues. The manuscript will benefit from subsections that separate these issues. Additionally, the manuscript will benefit from subsections separating the discussion of the overview and the discussion of the systematic review.

Throughout the manuscript subheadings have been added to separate the two issues regarding limitations to systematic reviews and limitations to primary studies. 

   2. “In a recent overview published in 2019, we conducted a thorough, systematic assessment of the general methodological quality of these systematic reviews using the AMSTAR criteria [19;20].”

Here, the authors should reference their overview. The authors should also clarify whether this overview addressed systematic reviews of processed meat only, red and processed meat, or general dietary exposures.

We have added the reference and clarified that the overview addressed systematic reviews of processed meat only.

  1. “More than 100 reviews were excluded, because they had not performed a quality assessment on the primary studies included in the review. In total, only 22 of 130 reviews and meta-analyses met these two basic criteria, and were subsequently included in our overview of reviews.”

Have these 22 reviews previously informed any dietary recommendations? It is possible that only the best systematic reviews are used to inform dietary recommendations. The authors should acknowledge this possibility.

 If feasible, the authors may consider using a sample of dietary guidelines by authoritative organizations and government bodies and examining the quality of systematic reviews that informed recommendations on red and processed meat.

Thank you for this idea. It would certainly be interesting to explore. However, since this would be an independent research question, it is beyond the scope of the present manuscript. Below we have tabulated the citation frequency in Scopus for the reviews we included that addressed cancer outcomes, and this may indicate that reviews that highlights positive/significant results are more frequently citated.     

Reference

Outcome

Citations in Scopus

(per October 2021)

Choi (2013)

Esophageal cancer

56

Zhu (2014)

Esophageal cancer

46

Li (2016)

Nasopharyngeal carcinoma

8

Zhao (2017)

Pancreatic cancer

26

Lou (2014)

Hepatocellular carcinoma 

38

Fang (2015)

Gastric cancer

100

Li (2012)

Gastric cancer

30

Zhao (2017)         

Gastric cancer

22

Zhu (2013)

Gastric cancer

67

Quach (2016)

Glioma

2

Saneei (2015)

Glioma

20

Wallin (2011)

Ovarian cancer

36

Solimini (2016)

Non-Hodgkin lymphoma

9

Yang (2015)

Non-Hodgkin lymphoma

14

Yang (2012)

Lung cancer

58

Xu (2014)

Oral cavity and orophanx cancer

28

Zhang (2017)

Renal cell carcinoma

5

Zhao (2017)

Colorectal cancer

52

O´Sulivan (2013)

Cancer mortality

95

Wang (2016)

Cancer mortality

181

 4. Two additional issues likely explain recommendations against processed meat. To date, few reviews only report relative effects and few reviews—if any—report absolute effects. While relative effects for red and processed meat may be positive and statistically significant, absolute effects are small. Further, dietary guidelines rarely, if ever, consider public values and preferences. If guideline panels considered values and preferences, they would likely realize that few participants would be willing to modify their consumption of red and processed meat based on such small and uncertain effects. A discussion of these two issues will strengthen the paper. See: Johnston BC, Guyatt GH. Causal inference, interpreting and communicating results on red and processed meat. Am J Clin Nutr. 2020 May 1;111(5):1107-1108.

Thank you for this suggestion and we fully agree. We have added this to the introduction on page 3, 3rd paragraph:

To date, few reviews only report relative effects of red and processed meat on cancer outcomes and few reviews—if any—report absolute effects. While relative effects for red and processed meat may be positive and statistically significant, absolute effects are small (less than 1 %) (reference). Further, dietary guidelines rarely, if ever, consider public values and preferences. Thus, while for sustainability and environmental concerns clearly reductions in meat consumption is advisable, public willingness to modify red and processed meat consumption may be less likely based on small and uncertain health effects (reference). 

Reference: Johnston BC, Guyatt GH. Causal inference, interpreting and communicating results on red and processed meat. Am J Clin Nutr. 2020 May 1;111(5):1107-1108.

Minor Issues

 1. “Until now, there have been few randomized trials that investigated the consumption of red meat and risk of colon cancer, as recently reviewed by Johnston and colleagues [16], and similarly, only two trials examining the effect of processed meat intake on cancer risk in multi-factorial designs that also including other dietary changes [10;17], both of which showed significant reductions in meat, without changes in cancer risk [10;17].”

 The trials that the authors describe as ‘multifactorial’ investigated the effects of interventions that only modified red meat consumption secondary to other characteristics of the diet. The Women’s Health Initiative tested a low-fat diet which resulted in reduced red meat consumption. The Polyp Prevention Trial tested a high fiber diet which also resulted in reduced red meat consumption. The trials were not factorial in design (defined as a trial that tests the effects of more than two interventions simultaneously). To avoid confusion with ‘factorial trials’, I suggest the authors avoid the term ‘multifactorial’ and instead describe that the interventions tested in the trials modified the diet of the participants in many ways, only one of which was red meat intake.

Thank you for this suggestion, which have been added and replaced “multifactorial”:

Similarly, only two trials examined the effect of different dietary patterns on cancer risk, and only one of which was red meat intake [10;17].

  1. “Our reason for rating down our certainty in these studies was due to the serious risk of bias (issues regarding confounding, missing data, risk of selection of the reported results were not sufficiently addressed), serious risk of imprecision due to wide confidence intervals, and serious risk of inconsistency due to unexplained variability between the included studies (so-called heterogeneity), while there were no worrying issues regarding indirectness or publication bias [18;37].”

 Please remove ‘risk of’ for imprecision and inconsistency.

“risk of” has now been removed for imprecision and inconsistency.

  1. “The issues on confounders are further complicated, as some of the statistical models include mediators, and/or colliders, and thus there is a high risk of over-adjustment bias, which may distort the result. For instance, red meat may increase weight gain and could act as a mediator that should not be controlled.”

 This is not a compelling argument because over adjustment would typically reduce observed associations whereas analyses addressing the association between processed meat and adverse health outcomes are typically positive and statistically significant.

The two sentences have now been removed from the manuscript.